# Newly Obtained Apple Pectin as an Adjunct to Irinotecan Therapy of Colorectal Cancer Reducing *E. coli* Adherence and β-Glucuronidase Activity

**DOI:** 10.3390/cancers13122952

**Published:** 2021-06-12

**Authors:** Anna Palko-Łabuz, Jerzy Maksymowicz, Beata Sobieszczańska, Agnieszka Wikiera, Magdalena Skonieczna, Olga Wesołowska, Kamila Środa-Pomianek

**Affiliations:** 1Department of Biophysics and Neuroscience, Wroclaw Medical University, 50-368 Wroclaw, Poland; anna.palko-labuz@umed.wroc.pl (A.P.-Ł.); jerzy.maksymowicz@student.umed.wroc.pl (J.M.); kamila.sroda-pomianek@umed.wroc.pl (K.Ś.-P.); 2Department of Microbiology, Wroclaw Medical University, 50-368 Wrocław, Poland; beata.sobieszczanska@umed.wroc.pl; 3Department of Biotechnology and General Technology of Foods, Faculty of Food Technology, University of Agriculture in Krakow, 30-149 Krakow, Poland; agnieszka.wikiera@urk.edu.pl; 4Department of Systems Biology and Engineering, The Silesian University of Technology, 44-100 Gliwice, Poland; magdalena.skonieczna@polsl.pl; 5Biotechnology Centre, Silesian University of Technology, 44-100 Gliwice, Poland

**Keywords:** colon cancer, pectin, irinotecan, apoptosis, inflammation, *E. coli*, bacterial β-glucuronidase, side-effects alleviation

## Abstract

**Simple Summary:**

Colorectal cancer (CRC) is the second cause of cancer death worldwide. Irinotecan is a drug widely used in CRC treatment. Unfortunately, colonic bacteria decompose the metabolite of irinotecan back to the active form of the drug resulting in severe side-effects of the treatment, such as diarrhea. The present work demonstrated that new apple pectin (PC) enhanced cytostatic action of irinotecan and, at the same time, reduced the activity of bacterial enzymes responsible for the appearance of side-effects in patients. Thus, novel pectin PC constitutes a promising candidate for an adjunct to irinotecan therapy that might alleviate its side-effects, increasing its therapeutic efficacy.

**Abstract:**

Colorectal cancer (CRC) is the second cause of cancer death worldwide. The composition and enzymatic activity of colonic microbiota can significantly affect the effectiveness of CRC chemotherapy. Irinotecan is a drug widely used to treat colon cancer. However, the transformation of a drug-glucuronide (SN-38G) back to its active form (SN-38) by bacterial β-glucuronidase (GUS) constitutes the primary reason for the observed intestinal toxicity of irinotecan. It was demonstrated that novel enzymatically extracted apple pectin (PC) might be a promising candidate for an adjunct to irinotecan therapy. PC itself reduced the viability of HCT 116 and Caco-2 colorectal cancer cells, induced apoptosis, and increased intracellular reactive oxygen species production. Moreover, PC enhanced the cytotoxic and proapoptotic effect of irinotecan (at concentrations below its IC_50_), i.e., synergistic effect was recorded. Additionally, PC exhibited potent anti-inflammatory properties and prevented adhesion of prototype adherent-invasive *E. coli* (AIEC) LF82 strain and laboratory K-12_C600_ strain to colon cancer cells. PC was also identified to be an effective inhibitor of bacterial GUS activity. Altogether, novel apple pectin was identified as a promising candidate for a supplement to irinotecan therapy that might alleviate its side-effects via inhibition of bacterial GUS and thus increasing its therapeutic efficacy.

## 1. Introduction

Colorectal cancer (CRC) takes the third position among the most frequently occurring cancers worldwide [1]. As reported by International Agency for Cancer Research of the World Health Organization (WHO) in 2020 more than 1.9 million new CRC cases were diagnosed and over 900,000 deaths were recorded [1]. In the great majority, CRC appears as a sporadic disease, typically diagnosed in patients over 50 and usually linked to environmental factors, such as western-type diet rich in red meat, fat, and sugar [2]. About 10% of CRC cases develop due to inherited syndromes, such as familial adenomatous polyposis and Lynch syndrome [2]. Moreover, the higher risk of CRC has been observed in patients with inflammatory bowel disease, such as ulcerative colitis and Crohn’s disease [3]. Local inflammation may modulate the pathogenesis of CRC, mostly via the complex network of inflammation mediators, such as cytokines, produced both by cancer cells themselves and by various types of cells constituting tumor microenvironment [4]. Additionally, the progression of CRC can be influenced by colonic microbiota [5,6] that is altered in cancer patients as compared to the healthy subjects [7,8]. The microbiota was also shown to modulate the response of tumor cells to chemotherapy [9,10].

The treatment options of CRC depend on the stage of disease. For locally advanced colon or rectal tumors concomitant combination of surgery and chemotherapy is recommended, while in metastatic CRC, chemotherapy appears to be preferable [11]. Irinotecan is a derivative of an alkaloid, camptothecin. Irinotecan hydrolysis by patients’ carboxylesterases yields its active form, SN-38, an inhibitor of topoisomerase I [12]. This drug is a common component of cytotoxic combinations recommended as a treatment for CRC patients, such as FOLFIRI (fluorouracil, leucovorin, and irinotecan) or FOLFOXIRI (fluorouracil, leucovorin, oxaliplatin, and irinotecan) [13,14]. Common side-effects of irinotecan include diarrhea, nausea, and vomiting [15,16]. SN-38 is metabolized via glucuronidation in the liver to irinotecan glucuronide (SN-38G) that enters the small intestine via biliary excretion [17]. Unfortunately, bacteria in the colon produce β-glucuronidase (GUS) that decomposes SN-38G back to its active form (SN-38), which is toxic for intestinal epithelium [18], causing the adverse effects of chemotherapy.

Pectin constitutes a significant component of dietary fiber. This structural acidic heteropolysaccharide is a component of plant cell walls [19]. The basic structural skeleton of pectin is a linear polymer of galacturonic acid (GalUA). In native pectin molecule, some distinct structural regions are present. In rhamnogalacturonan I (RG I), the backbone consists of repeating dimers of GalUA and rhamnose, and is substituted with side chains composed of galactose and arabinose. Sidechains of rhamnogalacturonan II (RG II) contain atypical, rare monosaccharides, including L-fucose, apiose, and acids: aceric, glucuronic, ketodeoxyoctonic (KDO), and 3-deoxy-lyxo-2-heptulopyranosylaric (DHA). Structural differences may depend on the plant source of the polysaccharide. Commercially available pectins are usually prepared from citrus peels or apple pomace by hot acid extraction [20]. Pectins extracted using this method are, in comparison with the native molecule, depleted in RG I and RG II that are considered to be crucial for prebiotic, immunomodulatory, and anticancer potency of pectins [21,22]. Recently, Wikiera et al. [23] developed an effective method for enzymatic isolation of apple pectins using limited doses of endo-cellulase and endo-xylanase. This procedure protects native molecules, especially its neutral sugars that are not damaged during isolation [23]. Pectin studied in the present work (PC) was enzymatically extracted from apple pomace using endo-cellulase from filamentous fungus *Trichoderma viride*. Its detailed structure was previously described [23], and its biological properties were partially characterized [24].

In the present work, we studied the influence of enzymatically obtained pectin, PC, on colon cancer cells and compared its properties to commercially available modified citrus pectin, SolC. It was assumed that PC is going to exhibit intrinsic anticancer activity. Next, due to presumed anticancer properties of PC, the enhancement of the anticancer activity of irinotecan (i.e., synergistic effect) was expected when the drug would be applied together with PC. Additionally, considering the crucial role of colonic microbiota in causing irinotecan-induced diarrhea, we supposed that enzymatically obtained pectin might alleviate side-effects of irinotecan both via inhibition of bacterial GUS in the colon or providing an additional substrate for the enzyme. The results presented below confirmed all of the above hypotheses.

## 2. Materials and Methods

### 2.1. Chemicals

SN-38 (7-ethyl-10-hydroxy-camptothecin), an active metabolite of irinotecan, was from Sigma-Aldrich (Poznan, Poland). PectaSol-C (SolC) low molecular weight modified citrus pectin with a documented antitumor activity [25] was purchased from ecoNugenics (Santa Rosa, C, USA). Stock solutions of pectins were prepared at a concentration of 10 mg/mL in double-distilled water. About 50 mM stock solution of SN-38 was prepared in dimethyl sulfoxide (DMSO) and stored at −20 °C. Stock solutions were diluted in a culture medium just before the experiments.

### 2.2. Pectin Isolation

Macromolecular apple pectin was extracted from dried apple pomace using cellulase (endo-β-1,4-glucanase, EC 3.2.1.4) produced by filamentous fungus Trichoderma viride (Sigma-Aldrich, Poznan, Poland, Cat. No. C9422). Isolation was performed with 50 U enzyme per 1 g of pomace, at pH 5.0, in 40 °C, for 10 h, according to a method described by Wikiera et al. 2016 [23]. The detailed characteristics of the obtained pectin are presented in the previous works [23,24].

### 2.3. Cell Culture

The experiments were performed on the human colon cancer cell lines HCT-116 and Caco-2 (obtained from ATCC collection). The cells were cultivated in medium DMEM-F12 (PAA) supplemented with fetal bovine serum (10% FBS for HCT 116 and 20% for Caco-2; Eurx) and 1% antibiotics (10,000 μg/mL streptomycin and 10,000 units/mL penicillin; Sigma-Aldrich (Poznan, Poland), at 37 °C in a humidified atmosphere with 5% CO_2_, under standard conditions. In addition, the medium for Caco-2 cells contained non-essential amino acid solution (1×) (Sigma-Aldrich, Poznan, Poland). The medium was changed twice a week. Confluent cultures were passaged using 0.25% trypsin. All experimental procedures were carried out in the log-phase of cell growth.

### 2.4. Cell Viability Assay

SRB assay. The sulforhodamine B (SRB, Sigma-Aldrich, Poznan, Poland) assay was performed, as previously described [26], with modifications. Briefly, 15,000 cells were seeded in 96-well plates and allowed for a 24-h attachment period (at 37 °C). Then, the cells were treated with the studied pectins (0.05–0.5 mg/mL with or without 5 nM SN-38) for 48 h. Control wells only contained medium. The further procedure was carried out as previously described [27]. Cytotoxicity of DMSO to the cells was found to be negligible. All experiments were repeated three times.

MTT assay. Cells were seeded on 96-well plates in the concentration of 150,000 cells/mL. About 24 h later, the medium was removed, and a fresh medium containing tested pectins (0.05–0.5 mg/mL with or without 5 nM SN-38) was added. The control wells contained DMSO diluted in a medium. The plates were incubated for 48 h at 37 °C in a humidified atmosphere with 5% CO_2_. Then, the medium was removed, and cells were washed with physiological saline (1 × PBS; Sigma-Aldrich, Poznan, Poland). In the next step of the experiment MTT (3-(4,5-dimethylthiazol-2-yl)-2,5-diphenyltetrazolium bromide) in the concentration of 0.5 mg/mL (diluted in RPMI 1640 without phenol red; Sigma-Aldrich, Poznan, Poland) was added to each well. The plates were incubated for at least 2 h yielding purple MTT formazan crystals in the wells. The incubation with MTT was followed by the dissolution of crystals in the mixture of acidic isopropanol (isopropanol:HCl, *v*/*v* 1:0.04). The absorbance of the obtained product was read at 570 nm using a microplate reader. Survival rate was expressed as the percentage of cell survival and calculated from the ratio (A_570_ of treated cells/A_570_ of control cells) × 100%. The experiments were repeated three times.

### 2.5. Isobolographic Analysis

By the CompuSyn software (ComboSyn Inc., Paramus, NJ, USA) according to the classic median-effect equation, as described by Chou and Martin, the combination index (CI) values were calculated [28].
(1)$$\mathrm{CI}=\frac{{\left(D\right)}_{1}}{{\left(\mathrm{Dx}\right)}_{1}}+\frac{{\left(D\right)}_{2}}{{\left(\mathrm{Dx}\right)}_{2}}$$
where: (Dx)_1_ is the dose of drug 1 alone that inhibits a system by x%, (Dx)_2_ is the dose of drug 2 alone that inhibits a system by x%, and (D)_1_ + (D)_2_ are the doses of drug 1 and 2 in combination that also inhibit a system by x%. CI values below 1 represent synergism, CI values equal to 1 indicate additive effect (i.e., no interaction), and CI values above 1 point to antagonism.

### 2.6. Flow Cytometry Analysis

In order to study the cell cycle, apoptosis, and intracellular level of reactive oxygen species (ROS), flow cytometry was applied. In these studies, the cells in concentration 150,000 cells/mL were seeded onto 12-well plates 24 h before the experiment. Then, the medium was removed, and a fresh medium containing the compounds (0.2 mg/mL pectins with or without 5 nM SN-38) was added. The plates were incubated for 48 h at 37 °C in a humidified atmosphere with 5% CO_2_.

The apoptosis was investigated using an Annexin-V apoptosis assay (BioLegend, San Diego, CA, USA) and propidium iodide (PI) solution (100 μg/mL; Sigma-Aldrich, Poznan, Poland). The cells were collected and centrifuged for 3 min at 235× *g*. After supernatant removal, the cells were washed with PBS and centrifuged once again. In the next step, the pellet of cells was suspended in cold Annexin-V binding buffer and stained with FITC-labeled Annexin-V and PI in the dark for 20 min at 37 °C. After this time, the final volume of Annexin-V binding buffer was added to the samples.

In order to analyze the cell cycle, collected cells were centrifuged for 3 min at 235× *g* (room temperature). After supernatant removal the cells were fixed with hypotonic buffer containing PI 100 µg/mL in PBS; 5 mg/l of citric acid; 1:9 Triton-X solution; RNase 100 µg/mL in PBS. The samples were incubated for 20 min (room temperature) in the dark and then stored on ice till they were measured.

The level of ROS in the cells treated with studied compounds was quantified using 2′,7′-dichlorofluorescein diacetate (DCFH-DA, Sigma-Aldrich, Poznan, Poland), a cell-permeable fluorogenic probe. The experiments were performed after 72 h of incubation with given compounds. The cells were harvested and centrifuged, washed with PBS, and centrifuged again. The pellet was resuspended in a culture medium and a cell-permeable non-fluorescent probe 2′,7′-dichlorofluorescein diacetate (DCFH-DA, Sigma-Aldrich, Poznan, Poland) in a final concentration of 30 µM was added. The samples were incubated in darkness for 30 min at 37 °C and then stored on ice till they were measured. DCFH-DA is hydrolyzed by cellular esterases into its polar form DCFH. Due to the intracellular ROS and other peroxides DCFH undergoes oxidation and turns into highly a fluorescent form 2′,7′-dichlorofluorescein (DCF).

Cytometric analyses were performed immediately using an Aria III flow cytometer (Becton Dickinson, Franklin Lakes, NJ, USA) with the FITC configuration (488 nm excitation; emission: LP mirror 503, BP filter 530/30) or PE configuration (547 nm excitation; emission: 585 nm) and at least 10,000 cells were counted.

### 2.7. Caspase-3 Activation Assay

The activity of caspase-3 was analyzed using a commercially available kit (GenScript Biotech, Leiden, the Netherlands). Cells were seeded (800,000 cells/well) onto a 6-well plate in 2 mL of medium and incubated for 24 h at 37 °C. Then, the cells were treated with PC or SolC alone (at 0.2 mg/mL), and in combination with SN-38 (5 nM). After incubation with studied compounds for 48 h cells were scraped and centrifuged (2000× *g*, 5 min, 25 °C). After cell lysis, spectrophotometric detection (A_405_) of the chromophore *p*-nitroanilide (pNA) was used to measure caspase-3 activity. The relative increase of caspase-3 activity was determined by calculating the ratio of the absorbance of pNA in the studied sample (treated with the compound) to the control (with no compound). Experiments were performed in triplicate.

### 2.8. Enzyme-Linked Immunosorbent Assay (ELISA)

HCT 116 cells were seeded in 96-well plates in the concentration of 150,000 cells/mL. The colorectal cancer cells were treated with 5 μM of lipopolysaccharide (LPS; trinitrophenol-lipopolysaccharide from *E. coli* O111:B4, Sigma-Aldrich, Poznan, Poland) added at the time of seeding. A fresh medium containing the compounds (0.2 mg/mL) was added after 24 h. After incubation with the studied compounds for 48 h interleukin 6 (IL-6) and cyclooxygenase 2 (COX-2) in the cancer cell culture lysates were quantified using ELISA (Human COX-2 ELISA Kit; Sigma-Aldrich, Poznan, Poland; IL-6 Human ELISA Kit-ThermoFisher Scientific, Waltham, MA, USA). ELISA was performed according to the manufacturer’s instructions. Each plate test was repeated three times.

### 2.9. E. coli Strains

Tests were performed on two reference strains. A prototype adherent-invasive *E. coli* (AIEC) LF82 (O83: H1) strain, kindly provided by Dr. Arlette Darfeuille-Michaud, Université d’Auvergne, France, and laboratory *E. coli* K-12_C600_ strain. *E. coli* strains were routinely cultured overnight in Luria broth (LB) with shaking at 37 °C.

### 2.10. Adherence Assay

Overnight *E. coli* cultures in LB medium were harvested and suspended in PBS to the optical density 6 × 10^8^ CFU/mL established spectrophotometrically at 600 nm. HCT 116 or Caco-2 cells were washed three times with pre-warmed PBS, and 200 µL of pectin solutions in PBS at concentrations 0.125, 0.25, and 0.5 mg/mL were added to the cells following 200 µL of *E. coli* suspensions. Cells untreated with pectins and infected with *E. coli* served as a positive control. Then the cells were incubated for two hours at 37 °C in a humid atmosphere with 5% CO_2_. At 2 h post-infection, cells were washed three times with PBS and lysed with 0.1% Triton X-100. Serial dilutions of bacterial lysates were plated onto nutrient agar and incubated overnight at 37 °C to count bacterial colonies (CFU). The assay was repeated three times in duplicate. The results are presented as the percentage of *E. coli* adhering to cells concerning negative control established to be 100%.

Wright-Giemsa staining. Intestinal epithelial cells were cultured on slides in the wells of a 24-well plate. After 2 h of *E. coli* adherence with the epithelial cells, the cells were washed three times and fixed with 4% formaldehyde for 10 min at room temperature. After washing off the formaldehyde, cells were stained with Wright-Giemsa for 30 min. The dye was rinsed off, and the slides were dried. Adhesion images were taken under a light microscope at 100× magnification.

### 2.11. E. coli Proliferation

The impact of apple pectins on *E. coli* proliferation was assessed by mixing 75 µL of *E. coli* suspensions in PBS at the optical density 6 × 10^8^ CFU/mL obtained from overnight cultures in LB with the equal volumes of pectins solutions in PBS at concentrations 0.125, 0.25, and 0.5 mg/mL. According to the adherence assay, the plate was incubated for two hours at 37 °C in an atmosphere with 5% CO_2_. The proliferation of *E. coli* was measured spectrophotometrically at 600 nm and compared to the growth of *E. coli* at time zero established to be 100%.

### 2.12. β-Glucuronidase (GUS) Activity Assay

Overnight *E. coli* LF82 and K-12_C600_ cultures in 3 mL of tryptic soy broth (TSB) were supplemented with 0.3 µM 4-nitrophenyl β-D-glucuronide (PNPG; Sigma-Aldrich, Poznan, Poland) one hour before the end of the incubation to induce GUS synthesis. Then, the cultures were centrifuged (16,000× *g*; 10 min; 4 °C). Bacterial cell pellets were washed twice in cold 100 mM sodium phosphate buffer (pH 6.8), followed by suspension in 1 mL of cold phosphate buffer, and mixing with 0.1 mm silica glass beads. The samples were then beaten in BeadBugTM microtube homogenizer five times for 30 sec each, with 30-sec incubations on ice in between. The tubes were centrifuged for 10 min, 16,000× *g* at 4 °C, and bacterial extracts were collected and stored on ice. Bacterial extracts were prewarmed at 37 °C, and samples of 75 µL were mixed in microtitrate plate with prewarmed equal volumes of the 10 mM PNPG in phosphate buffer. The yellow 4-nitrophenol product was recorded in a spectrophotometer after 60 min of incubation at 37 °C at 405 nm. The 4-nitrophenol concentration was determined from a standard curve according to Aich et al. [29]. Protein concentrations in bacterial extracts were determined with a bicinchoninic acid kit (Pierce, Waltham, MA, USA). GUS activity was calculated as follows: 4-nitrophenol concentration/100 mg/mL protein × 60 min. The effect of pectins SolC and PC on *E. coli* GUS activity was assayed as described above. SolC and PC solutions in PBS at final concentrations of 0.25, 0.5, and 0.7 mg/mL were mixed with an appropriate amount of *E. coli* lysates and 10 mM PNPG in the wells of a 96-well plate and incubated at 37 °C for 18 h.

### 2.13. Statistical Analysis

All of the experiments were repeated three times. Data represent the mean ± standard deviation (SD) of at least three replications. The Student’s *t*-test was applied and *p*-values less than 0.05 were considered to achieve statistical significance.

## 3. Results

### 3.1. Anticancer Activity of Pectins

#### 3.1.1. Cytotoxicity of Pectins

The influence of two pectins on colon cancer cell viability was assessed. Commercially available pectin PectaSol-C (SolC) was compared to PC that had been enzymatically extracted from apple pomace using endo-cellulase. SolC is derived from water-insoluble citrus pectin modified by pH/temperature treatment to increase its water solubility and anticancer properties [25].

Both pectins were revealed to possess anticancer activity against human colorectal carcinoma HCT 116 cells (Figure 1A,B), as shown by the results of SRB as well as MTT assays. In both types of assays, PC exhibited significantly greater cytotoxic activity as compared to SolC (*p* < 0.05 for all concentrations tested). IC_50_ values obtained for PC were 0.098 ± 0.011 mg/mL and 0.103 ± 0.014 mg/mL for SRB and MTT assay, respectively. In the case of SolC, the SRB assay yielded an IC_50_ value of 0.590 ± 0.019 mg/mL, whereas the MTT assay gave IC_50_ equal to 0.640 ± 0.018 mg/mL. The appearance of HCT 116 cells cultures treated with 0.2 mg/mL of pectins is presented in Figure 1E.

Cytotoxic activity of the studied pectins was also studied in human colorectal adenocarcinoma cells Caco-2 (Appendix A). The results of SRB and MTT cell viability assays revealed that PC was more toxic to Caco-2 cells than SolC, similarly as in the case of HCT 116 cells. IC_50_ values calculated for SolC were 0.440 ± 0.016 mg/mL (SRB) and 0.490 ± 0.015 mg/mL (MTT) that were slightly lower than that obtained in HCT 116 cells. On the other hand, IC_50_ values obtained for PC in Caco-2 cells, 0.210 ± 0.009 mg/mL for SRB and 0.270 ± 0.013 mg/mL for MTT, were almost twice higher than in HCT 116 cells. Therefore, it was concluded that the difference in cytotoxicity between SolC and PC was more pronounced in the case of HCT 116 cell line than in Caco-2.

#### 3.1.2. Cytotoxicity of Pectins Combined with SN-38

Additionally, the anticancer activity of pectins was studied in the presence of SN-38, an active form of irinotecan—the drug commonly used to treat colon cancer. Its IC_50_ value in HCT 116 cell line was found to be 10.2 ± 0.8 nM (3.5 x 10^−6^ mg/mL). Figure 1E presents the appearance of cell cultures treated with 5 and 50 nM of SN-38 as well as with the mixtures of 0.2 mg/mL of pectin and 5 nM of SN-38. It can be noticed that in the case of PC:SN-38 combination the number of HCT 116 cells is reduced to a greater degree than in case of the treatment with SN-38 (at 5 nM) alone. Figure 1C,D present the results of the SRB and MTT assay, respectively. Cell viability was significantly reduced in the presence of both 0.2 mg/mL of SolC and PC. In addition, the combination of pectins with a low concentration of SN-38 reduced cell viability even more. The isobolographic analysis was employed to evaluate the possible synergy between the anticancer drug and pectins. Dose and effect data obtained from SRB assay for pure compounds and for two-component combinations (pectin:SN-38) were subjected to CompuSyn analysis (Table 1). The cytotoxic effects of PC and SN-38 were synergistic as demonstrated by the combination index (CI) values well below 1. On the other hand, no synergy was detected between SolC and SN-38, their effect was purely additive.

### 3.2. Proapoptotic Activity of Pectins

#### 3.2.1. Annexin V/PI Double Staining Assay

After confirming the anticancer properties of SolC and PC, it was checked whether the treatment of colon cancer cells with the studied pectins led to cellular death by apoptosis. Therefore, Annexin V/PI double staining assay was performed in HCT 116 cells. Typical dot plots obtained during the assay are presented in Appendix A. As shown in Figure 2, PC at 0.2 mg/mL slightly increased the number of apoptotic cells as compared to the control not treated by the pectin. Such an effect was not observed for SolC tested at the same concentration. The application of SN-38 also resulted in a significant increase of apoptotic cell population, both at 5 and 50 nM concentration. The drug when applied in the highest concentration also caused necrosis of HCT 116 cells. The apoptosis-inducing activity of pectins was also tested in combination with SN-38 at the concentration of 5 nM. It was observed (Figure 2B) that in the case of PC the number of cells undergoing apoptosis was much bigger as compared to the probe treated with SN-38 only (*p* < 0.05). On the other hand, the combined SolC:SN-38 treatment resulted in lower number of apoptotic cells as compared to the probe treated with SN-38 alone (*p* < 0.05). It was concluded that PC augmented the proapoptotic activity of SN-38.

#### 3.2.2. Caspase-3 Activation

Additionally, the ability of pectins to activate the apoptosis-involved protease, caspase-3, was investigated (Figure 3). Caspase-3 is an enzyme associated with the initiation of the apoptotic cascade. Its activation ultimately leads to the execution of the program of cell death. Both studied pectins activated this enzyme in concentrations higher than 0.1 mg/mL. The addition of 5 nM of SN-38 to the cells also treated with 0.2 mg/mL of pectin significantly enhanced the activity of caspase-3. SN-38 used alone at 5 nM did not, however, cause any activation of caspase-3. This experiment additionally confirmed the proapoptotic properties of the studied pectins and demonstrated that PC enhanced the ability of SN-38 (applied in concentration below IC_50_) to induce cell death.

### 3.3. The Influence of Pectins on Cell Cycle

The influence of pectins on the cell cycle of HCT 116 cells was also tested (see Figure 4 and Appendix A for the histograms). The cytometric evaluation indicated that both pectins at the concentration of 0.2 mg/mL elevated sub-G_1_ cellular fraction (dead cells), with PC being significantly more active than SolC. The number of dead cells was also elevated when HCT 116 cells were treated with SN-38, the higher concentration of the drug inevitably causing the more pronounced effect. The most interesting observation was made for the PC:SN-38 mixture. When the cells were treated by the two compounds in combination, the very strong increase of sub-G_1_ cellular fraction occurred that was accompanied by the reduction of the abundance of all other fractions. On this basis, it was concluded that PC enhanced the cytotoxic potential of SN-38.

### 3.4. The Effect of Pectins on Reactive Oxygen Species (ROS) Production

The assay based on the detection of intracellular ROS by the fluorescent probe DCF was applied to quantify the generation of ROS in the presence of the studied pectins. As shown in Figure 5. the treatment of HCT 116 cells with SolC at 0.2 mg/mL concentration resulted in a slight decrease in ROS level, whereas PC applied at the same concentration elevated the production of ROS by cancer cells. SN-38 (at 5 nM) gently increased ROS level. When the anticancer drug was combined with pectins, the increase in ROS generation was observed, by more than 60% in the case of SolC and almost 4.5-fold in the case of PC. The conclusion was drawn that SolC itself possessed some antioxidative properties, in contrast to PC that was prooxidative. On the other hand, both pectins strengthened ROS production induced by SN-38.

### 3.5. Anti-Inflammatory Activity of Pectins

To check whether the studied pectins could modify the cellular inflammatory response HCT 116 cells were pretreated by 5 μM of bacterial lipopolysaccharide (LPS, endotoxin) for 24 h before the experiment to induce inflammation. After the treatment with pectins the amounts of cyclooxygenase 2 (COX-2) and interleukin 6 (IL-6) were quantified in cell culture lysates. As presented in Figure 6 in not-stimulated cells pectins tested at 0.2 mg/mL had no influence on either COX-2 or IL-6 levels. As expected, the treatment of colon cancer cells by LPS induced inflammatory response resulting in the substantial increase of COX-2 and IL-6 produced by the cells (Figure 6A and Figure 6B, respectively). The addition of any of the studied pectins resulted in the reduction of LPS-induced levels of both inflammation markers. The effect of PC was more pronounced than that of SolC (statistical significance (*p* < 0.05) was obtained both in the case of COX-2 and IL-6). It was demonstrated that both studied pectins were able to modulate the inflammatory response of colon cancer cells.

### 3.6. Adherence of E. coli to Colon Cancer Cells

Since colonic microbiota is known to affect colorectal cancer initiation and progression, adherence of bacteria *E. coli* to cancer cells in the presence of the studied pectins was monitored. Two bacterial strains were employed, a laboratory *E. coli* K-12_C600_ strain, and an adherent-invasive *E. coli* (AIEC) LF82 strain. The latter was isolated from a patient with Crohn’s disease. SolC did not influence the ability of either LF82 (Figure 7A) or K-12_C600_ (Figure 7B) strains to adhere to HCT 116 cells. On the other hand, PC in concentrations 0.25 mg/mL or higher significantly reduced the adherence of both *E. coli* strains to cancer cells. The observed effect seemed slightly stronger in AIEC *E. coli* than in the control laboratory strain. It was also confirmed that none of the studied pectins influenced cancer cell viability (Appendix A) and *E. coli* proliferation (Appendix A) under experimental conditions. The same experiment was performed in human colorectal adenocarcinoma cells Caco-2 (Appendix A), and it showed virtually the same results. Representative pictures of *E. coli* LF82 strain adhering to untreated Caco-2 cells and in the presence of PC at different concentrations are shown in Figure 8. It was therefore found that PC was able to reduce adherence of *E. coli* to colon cancer cells.

### 3.7. The Effect of Pectins on β-Glucuronidase (GUS) Activity

Colonic bacteria may influence intestinal toxicity of irinotecan by enzymatic decomposition of SN-38G back to highly cytotoxic SN-38. The influence of pectins on the activity of β-glucuronidase produced by *E. coli* strains was tested. As shown in Figure 9, both studied pectins reduced GUS activity in a concentration-dependent manner. PC exhibited much stronger enzyme-inhibiting potency as compared to SolC. It was concluded that pectins might potentially alleviate irinotecan-induced diarrhea by reducing enzymatic transformation of the drug in the colon.

## 4. Discussion

In the present work, it was demonstrated that both studied pectins, PC and SolC, significantly reduced the viability of two colon cancer cell lines, HCT 116 and Caco-2. In both types of cells, enzymatically isolated apple pectin (PC) exhibited greater cytotoxicity in comparison to commercially available modified citrus pectin (SolC). Previous studies on PC revealed its ability to reduce proliferation, cell adhesion, and invasion of human adenocarcinoma HT-29 and melanoma B16F10 cells [24]. Pectin treatment was much more destructive to colon cancer cells than to melanoma cells. At the same time, it was shown that PC (tested in concentrations up to 1 mg/mL) exerted virtually no effect on proliferation and adhesion of normal mouse fibroblasts L929 [24]. Anticancer activity of various pectins has been already recognized for several years (see [21,30] for a review). Pectins isolated from ginseng [31] and potato [32] were demonstrated to exert an antiproliferative effect on colon cancer cells (HT-29). Interestingly, it seems that anticancer activity of commercial pectin preparations was limited while enzymatically modified pectins, especially the ones rich in RG I regions, significantly reduced the proliferation of various cancer cell lines [33,34].

Moreover, PC but not SolC was demonstrated to act synergistically with SN-38 in reducing colon cancer cells viability. Pectins have not been reported to potentiate the activity of this drug before. In contrast, the augmentation of the antiproliferative effect of irinotecan toward HCT 116 cells was observed in case of flavopiridol [35]. On the other hand, the synergistic anticancer effect of modified citrus pectin preparation and paclitaxel [36], as well as doxorubicin [37] was recorded in ovarian and prostate cancer cell lines, respectively.

To understand better the mechanism of anticancer properties of pectins, their ability to induce apoptosis in colon cancer cells was studied. It was shown that PC but not SolC significantly increased the size of the apoptotic cell population. Both pectins activated caspase-3, an effector caspase in the apoptotic pathway. Cell cycle analysis demonstrated both pectins elevated sub-G_1_ cellular fraction that is composed of cellular debris as well as late apoptotic and necrotic cells. SN-38, tested at low concentrations, also induced apoptosis in HCT 116 cells and raised the abundance of sub-G_1_ cellular fraction but it did not activate caspase-3. The most interesting findings came from the analysis of HCT 116 cells treated with PC and SN-38 simultaneously. Strong increases of apoptotic cell population, as well as sub-G_1_ cellular fraction, suggested that PC augmented proapoptotic and cytostatic properties of this anticancer drug. Proapoptotic activity of pectins have been already demonstrated for many types of pectin preparations in a variety of experimental settings, e.g., in ovarian cancer (SolC [36]), prostate cancer (fractionated pectin powder [38], SolC [39]) as well as in colon cancer (pectic oligosaccharides [40], sweet potato [41], ginger [31], and beetroot pectin [33]). Modified citrus pectin was recorded to increase caspase-3 activity and to modify the cell cycle of ovarian cancer cells in a similar way as observed for apple pectin PC in the present work [36]. In the same study, the synergy of pectin and paclitaxel in apoptosis induction was observed. However, the same pectin preparation induced G_2_/M phase arrest, activation of caspase-3, and the cleavage of poly (ADP-ribose) polymerase (PARP) in urinary bladder cancer cells [42]. Since dietary pectins are not digested before they reach the distal portion of the gastrointestinal tract, they cannot reach circulation. Pectins are known to bind strongly to the mucous layer of the intestine, and in this way, they can modulate the function of epithelial cells. Avivi-Green et al. [43] fed rats in which carcinogenesis had been chemically induced a high-pectin diet and observed an increased apoptotic index of their colon crypt cells that was likely to result in reduced tumor volumes in comparison with rats fed with a standard diet. It has been presumed that proapoptotic, and more generally, anticancer activity of pectins was related to their interaction with galectin-3 (Gal-3). Gal-3 is a lectin that specifically binds β-galactosides. It is present in a variety of cells and often over-expressed in cancer (see [44,45] for a review). Gal-3 specifically binds many intra- and extracellular proteins mediating numerous biological processes including the ones necessary for cancer growth, transformation, invasion, and metastasis. The studies trying to identify the structural features of pectin needed for proapoptotic activity pointed to the importance of an ester-based cross-link within the pectin molecule [38] as well as to the high content RG-I regions (rich in neutral sugars) within the pectin preparation [33].

The proapoptotic properties of chemical compounds are often associated with their ability to increase the level of reactive oxygen species (ROS) within cells. This possibility was tested in the case of the studied pectins. It was found that PC but not SolC (both at 0.2 mg/mL) increased ROS content in colon cancer cells. SN-38, when applied alone at the concentration of 5 nM increased ROS level only slightly. The SN-38-induced increase in ROS content was pretty elevated when the drug was combined with SolC and considerably raised in the case of the PC:SN-38 combination. Enzymatically isolated pectin, PC, has been previously demonstrated to effectively scavenge free radicals, and its antioxidative potency was more significant as compared to commercial apple pectin [24]. Antioxidative activity of various types of pectins has been widely reported (see [46] for a review). However, it should be taken into consideration that the laboratory-detected ability of a compound to scavenge free radicals does not have to translate into its ROS-mitigating behavior in the cellular environment. Modified citrus pectin has been recently demonstrated to increase ROS production in prostate cancer cells that contributed to their increased radiosensitivity in the presence of pectin [47]. Both citrus and apple pectins were observed to increase ROS release in breast cancer cells and to induce apoptosis via the mitochondrial pathway [48].

Local inflammation in the colon has been associated with a higher risk of CRC development [5]. Therefore, it was decided to investigate the ability of the studied pectins to modulate the inflammatory response of colon cancer cells. Since HCT 116 cells express Toll-Like Receptor 4 (TLR4) [49,50], LPS produced by *E. coli* was used to induce inflammatory processes in these cells. Interaction of TLR4 with LPS switches on the signaling pathway in which many components (including nuclear factor NFκB) are engaged and that triggers the expression of pro-inflammatory cytokines (e.g., IL-6) and enzymes, like COX-2 [51,52]. The results showed that LPS induced a significant increase in both IL-6 and COX-2 levels produced by HCT 116 cells. In the presence of pectins the levels of both proteins significantly decreased. The immunomodulatory effect was, however, much more pronounced in the case of PC than SolC. The interplay of pectins with various components of LPS/TLR4-induced pathways has already been observed. Citrus pectin caused the decrease of expression of inducible nitric oxide synthase (iNOS) and COX-2 in LPS-activated macrophages [53]. An apple oligogalactan suppressed both the expression and activity of COX-2 in LPS-stimulated colon carcinoma cells [54]. In the same experimental setting modified apple polysaccharides significantly decreased LPS-induced expression of TLR4, COX-2, matrix metallopeptidases (MMP) 2 and 9, iNOS, as well as prostaglandin E2 [55]. Oral application of pectins was also demonstrated to alleviate LPS-induced shock in mice [56] and to decrease the expression of pro-inflammatory markers in model mouse colitis [57]. The direct interaction of pectins with TLR4, thus competing with LPS binding, was proposed to constitute the molecular basis for the modulation of the inflammatory process by pectins [55,58]. Interestingly, the sugar profile and structure of RG II region present in PC molecule (but not in SolC) highly resemble the structure of O-antigen of LPS [59,60].

Pectins are characterized by high adhesion to the intestinal mucous layer. They can both affect the function of epithelial cells and form the protective barrier against colonization by opportunistic microflora [61]. Additionally, they form a favorable environment for the normal microbiome and constitute the source of nutrients for bacteria populating the colon [62]. The disruption of gut microbiome equilibrium, called dysbiosis, often precedes the development of certain pathological conditions, from mild as irritable bowel syndrome to severe such as CRC. Colonic microbiota of CRC patients has been shown to differ from healthy individuals colonic flora [8,9]. Therefore, the impact of PC and SolC on the adherence of *E. coli* to colorectal cancer cells was studied. Two bacterial strains have been applied, laboratory K-12_C600_ strain, and adherent-invasive *E. coli* (AIEC) LF82 strain, isolated from a Crohn’s disease patient. SolC did not change the adherence of *E. coli* to HCT 116 and Caco-2 cells, whereas PC significantly diminished the adherence of bacteria to colon cancer cells. Moreover, the inhibitory action of PC seemed to be stronger in AIEC *E. coli* than in laboratory strain. The inhibitory effect of pectic oligosaccharides (POS) on the adhesion of pathogenic bacterial strains has been demonstrated in numerous studies. In human colon adenocarcinoma epithelial cells HT-29 POS from citrus significantly reduced the adherence of *Vibrio cholerae* [63], several enteropathogenic as well as verotoxigenic *E. coli* strains [64], and Shiga-toxin-producing *E. coli* O157:H7 [65]. Although orange peel POS did not affect *Campylobacter jejuni* adhesion, they reduced its invasion into intestinal epithelial cells Caco-2 [66]. In the study on POS produced from apple pomace, Wilkowska et al. [67] presented evidence for their inhibitory effect on adhesion of pathogenic strains of *E. coli*, *Listeria monocytogenes*, and *Salmonella typhimurium* to Caco- 2 cells. In contrast, the adhesion of probiotic bacterial strains to the intestine epithelium was increased in the presence of pectins. Similarly, pectin-like oligosaccharides from *Panax ginseng* roots was shown to reduce adhesion of *Helicobacter pylori Actinobacillus actinomycetemcomitans, Propionibacterium acnes*, and *Staphylococcus aureus* to human gastric adenocarcinoma cells with no inhibitory effects against non-pathogenic *Lactobacillus acidophilus, E. coli*, and *Staphylococcus epidermidis* [68]. These studies pointed to the beneficial effects of pectins in maintaining healthy microbiota.

The composition and enzymatic activity of colonic microbiota can significantly affect the effectiveness of chemotherapy [10,11]. On the other hand, chemotherapeutic drugs can influence intestinal microbiota [69]. Decomposition of a drug-glucuronide back to an active form of the drug by bacterial β-glucuronidase constitutes the major reason for the observed intestinal toxicity of irinotecan [18] but also other xenobiotics [70]. The application of GUS inhibitors as an adjuvant to irinotecan therapy has been proposed and demonstrated to be effective in alleviating the drug-induced gut damage and diarrhea [71,72]. For this reason, it was decided to investigate the effect of the studied pectins on the activity of GUS produced by two *E. coli* strains, laboratory strain and the one isolated of Crohn’s disease patient. Both SolC and PC were demonstrated to inhibit GUS activity. However, the reduction caused by PC was much stronger as compared to SolC. The analysis of previous studies dealing with the influence of pectins on bacterial GUS revealed lots of inconsistencies. Depending on the type of pectin and experimental setting pectins were reported to increase [73,74], to have no effect [75,76,77], or to reduce GUS activity [78,79,80,81]. Based on the results obtained in the present study, it was concluded that PC strongly inhibited the enzymatic activity of *E. coli* GUS and might therefore reduce the backward transformation of SN-38G to SN-38.

## 5. Conclusions

The novel pectin, PC, obtained by enzymatic extraction of apple pomace, demonstrated to possess superior anticancer activity as compared to the modified citrus pectin, SolC. The key findings of the present study are summarized in Figure 10. PC reduced colon cancer cell viability, induced apoptosis, and increased intracellular ROS production. Moreover, the enhancement of the cytotoxic and proapoptotic effects of low concentrations of irinotecan was observed in the presence of PC. Additionally, novel apple pectin exhibited potent anti-inflammatory properties, prevented adhesion of *E. coli* to colon cancer cells, as well as inhibited the activity of bacterial GUS. With the reservation that the conclusions are based on the in vitro study only, it was demonstrated that novel pectin might be a promising candidate for an adjunct to irinotecan therapy that might alleviate its side effects and thus increase its therapeutic efficacy.

## Figures and Tables

**Figure 1 cancers-13-02952-f001:**
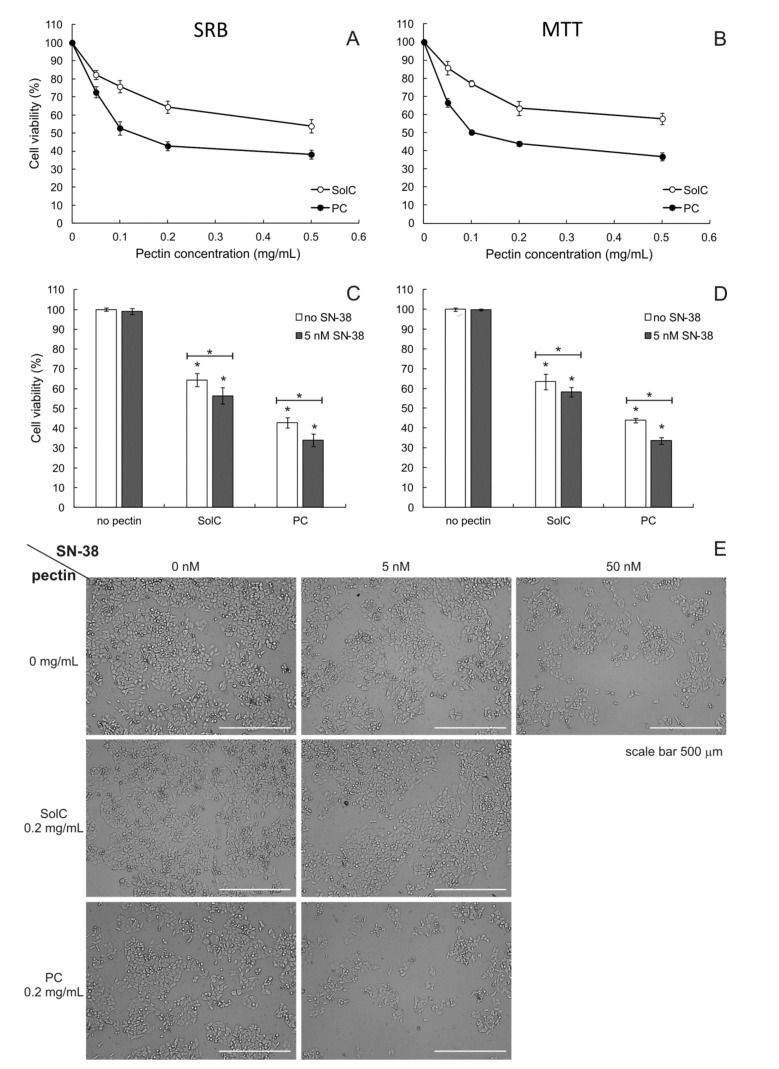
SRB (**A**) and MTT (**B**) cytotoxicity assay of pectins and 0.2 mg/mL pectins in combination with SN-38 (**C**,**D**) for SRB and MTT, respectively) in HCT 116 cells (incubation time 48 h). Light microscopy images (magnification 10×) of HCT 116 cells (**E**) treated with pectins (0.2 mg/mL) and/or 5 nM SN-38. The means of three experiments ± SD are presented (* *p* < 0.05). Statistical significance was checked between the studied probes and controls (no pectin) and between probes containing only pectin and pectin combined with SN-38. In panels A and B all measurements were significantly different from the control (not marked on the graph for the sake of clarity).

**Figure 2 cancers-13-02952-f002:**
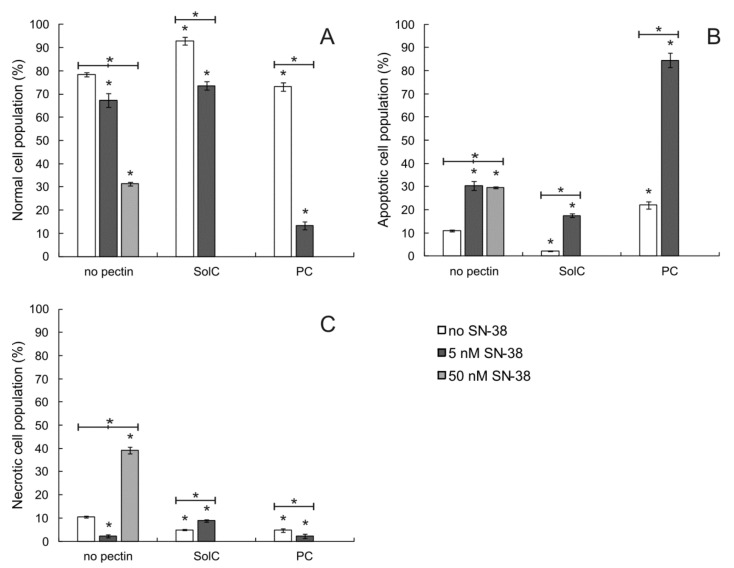
The proportion of normal (**A**), apoptotic (**B**), and necrotic (**C**) cell populations as recorded by Annexin-V apoptosis assay in HCT 116 cells treated with pectins (0.2 mg/mL) and/or SN-38 for 48 h. The means of three experiments ± SD are presented (* *p* < 0.05). Statistical significance was checked between the studied probes and controls (no pectin), and between probes containing only pectin and pectin combined with SN-38. Cells were recognized as viable (Annexin-V and PI negative), apoptotic (Annexin-V positive and PI negative), and necrotic (Annexin-V and PI positive) based on the measurement of cell-associated fluorescence of FITC-Annexin-V conjugate and PI.

**Figure 3 cancers-13-02952-f003:**
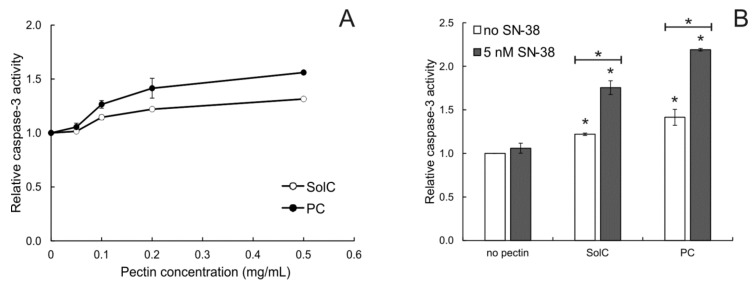
Relative caspase-3 activity in HCT 116 cells treated with pectins (**A**) and 0.2 mg/mL pectins in combination with SN-38. (**B**) for 48 h. The means of three experiments ± SD are presented (* *p* < 0.05). Statistical significance was checked between the studied probes and controls (no pectin), and between probes containing only pectin and pectin combined with SN-38. In panel A only the measurements for 0.05 mg/mL of SolC and PC were not significantly different from the control (not marked on the graph for the sake of clarity).

**Figure 4 cancers-13-02952-f004:**
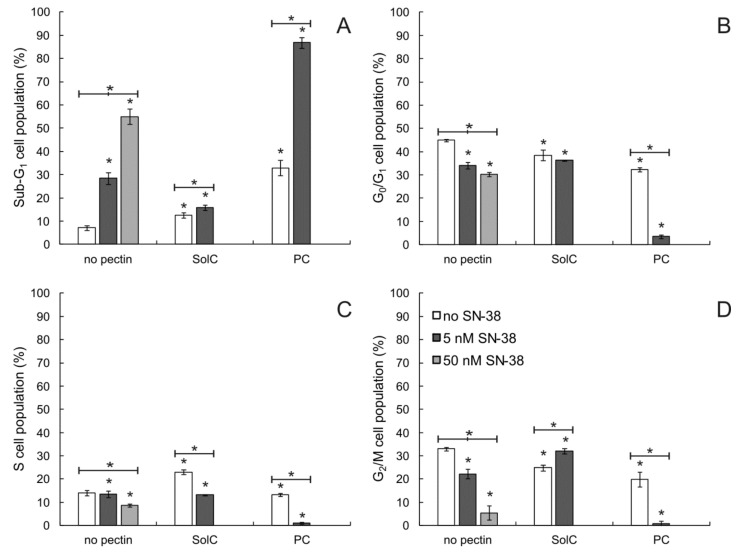
Cell cycle-dependent DNA content in HCT 116 cells treated with 0.2 mg/mL of pectins and /or SN-38 for 48 h. Sub-G1 population (**A**)—dead cells, G0/G1 (**B**)—mononuclear cells, S (**C**)—DNA replication, G2/M (**D**)—mitosis. Statistical significance was checked between the studied probes and controls (no pectin) as well as between probes containing only pectin and pectin combined with SN-38. * *p* < 0.05.

**Figure 5 cancers-13-02952-f005:**
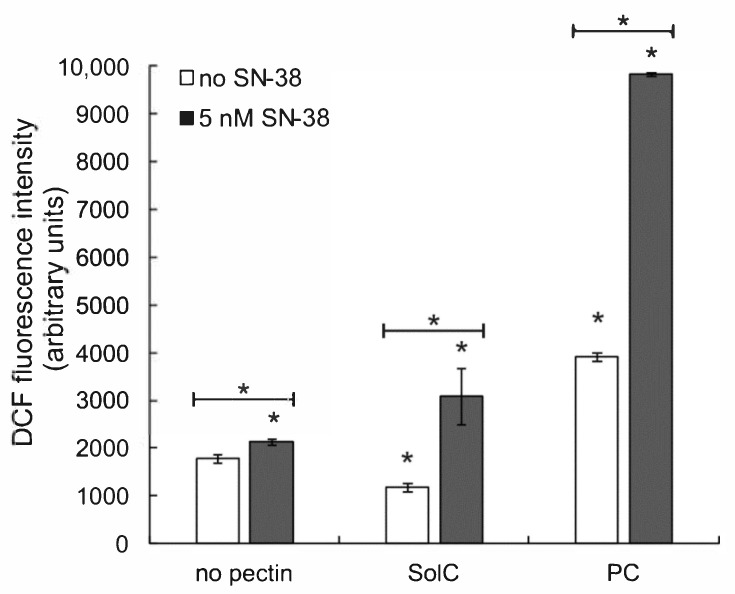
ROS level in HCT 116 cells treated with 0.2 mg/mL pectins and/or 5 nM SN-38 for 48 h. The means of three experiments ± SD are presented (* *p* < 0.05). Statistical significance was checked between the studied probes and controls (no pectin) as well as between probes containing only pectin and pectin combined with SN-38.

**Figure 6 cancers-13-02952-f006:**
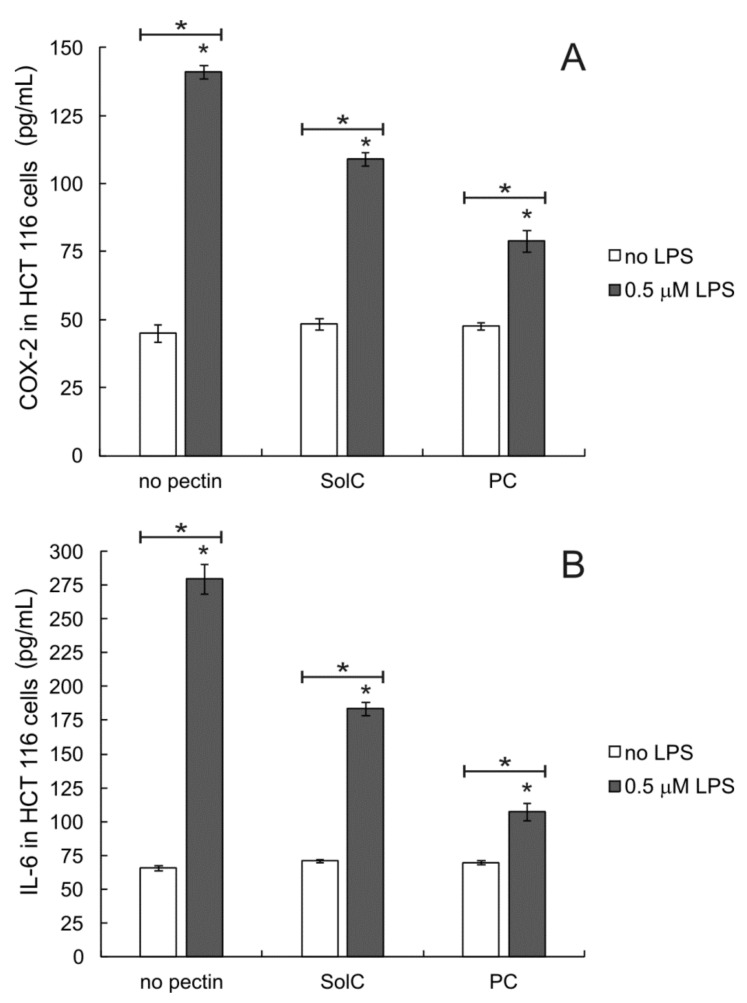
Amount of COX-2 (**A**) and IL-6 (**B**) in HCT 116 cells treated with 0.2 mg/mL pectins and/or 0.5 μM LPS. Cells were pretreated with LPS for 24 h and then incubated with pectins for 48 h. The means of three experiments ± SD are presented (* *p* < 0.05). Statistical significance was checked between the studied probes and controls (no pectin) as well as between probes containing only pectin and pectin combined with LPS.

**Figure 7 cancers-13-02952-f007:**
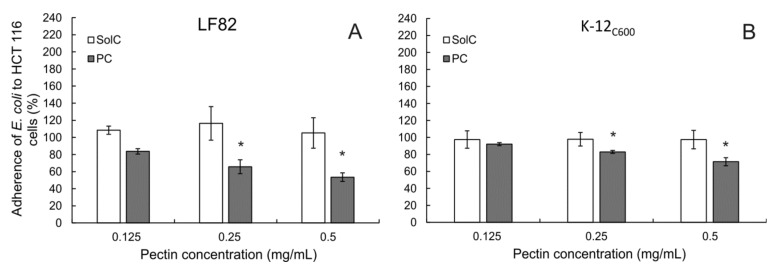
*E. coli* LH82 (**A**) and K-12_C600_ strains (**B**) adherence to HCT 116 cells in the presence of pectins during 2 h of incubation. The means of three experiments ± SD are presented. Statistical significance (* *p* < 0.05) was checked between the studied probes and controls (no pectin) assumed to be 100%.

**Figure 8 cancers-13-02952-f008:**
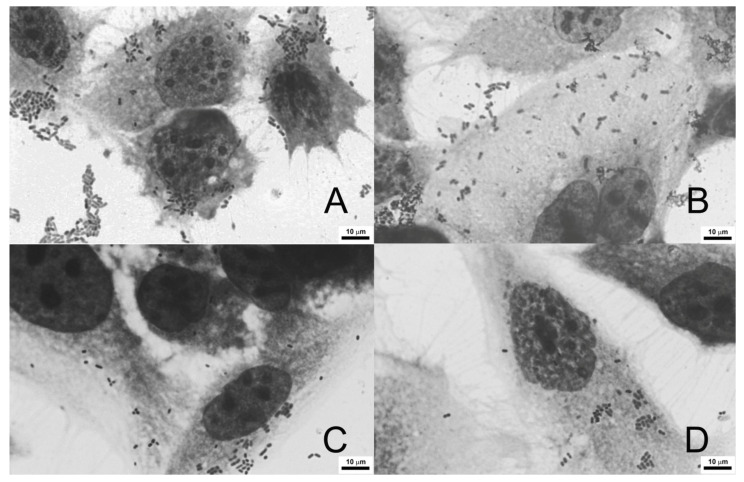
Adherence of *E. coli* (LF82 strain) to Caco-2 cells. Representative pictures of bacteria adhering to untreated Caco-2 cells (**A**), and treated with PC at concentration of 0.125 mg/mL (**B**), 0.25 mg/mL (**C**), and 0.5 mg/mL, (**D**). Wright-Giemsa stain, 100× magnification.

**Figure 9 cancers-13-02952-f009:**
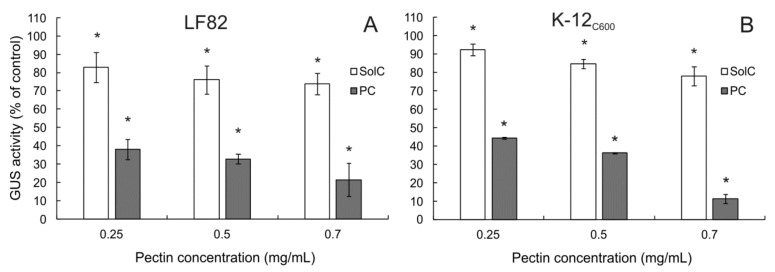
The influence of pectins on GUS activity in *E. coli* LH82 (**A**) and K-12_C600_ lysates (**B**). Pectin solution were mixed with 0.5 U of *E. coli* GUS and incubated for 18 h. The means of three experiments ± SD are presented (* *p* < 0.05). Statistical significance was checked between the studied probes and controls (no pectin) assumed to be 100%.

**Figure 10 cancers-13-02952-f010:**
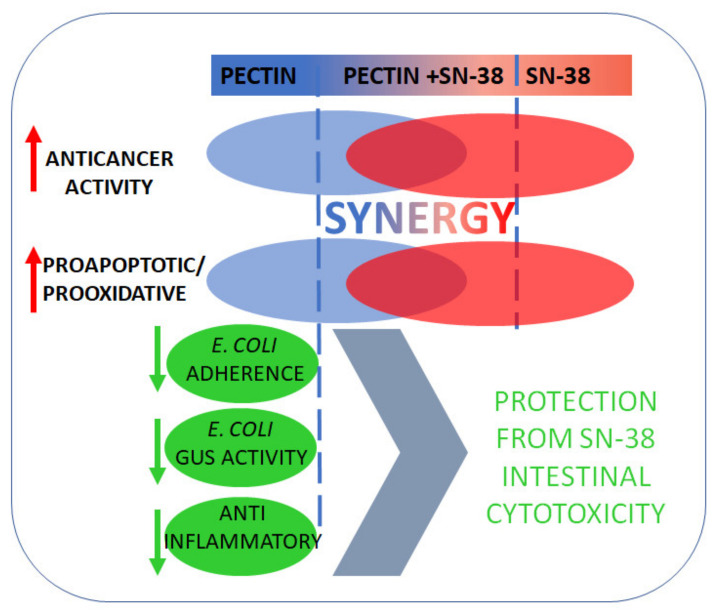
Schematic representation of activities of novel apple pectin in colon cancer cells.

**Table 1 cancers-13-02952-t001:** Combination of pectins with SN-38 against HCT 116 cell growth.

Concentration (mg/mL)		Ratio	Combination Index (CI)
SN-38	SolC		
1.96 × 10^−6^	0.2	102,000:1	1000
**SN-38**	**PC**		
0.96 x 10^−6^	0.1	104,400:1	0.822
1.96 x 10^−6^	0.2	102,000:1	0.779
3.92 x 10^−6^	0.5	127,550:1	0.792

Dose and effect data were obtained from the sulforhodamine B (SRB) assay (mean values of three experiments) and analyzed by CompuSyn software (ComboSyn Inc., Paramus, NJ, USA). CI values were calculated by CompuSyn software. CI = 1 indicates additive effect, CI < 1—synergism, and CI > 1—antagonism.

## Data Availability

Data sharing is not applicable to this article.

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
