# Peer review of "Newly Obtained Apple Pectin as an Adjunct to Irinotecan Therapy of Colorectal Cancer Reducing E. coli Adherence and β-Glucuronidase Activity"

_cancers, 2021, doi:10.3390/cancers13122952_

Round 1

Reviewer 1 Report

In this study, Palko-Labuz et al examined the impact of apple pectin on the treatment of colon cancer cell lines with irinotecan. The study has a strong clinical and scientific rationale and the study team have demonstrated a great effort to internally validate their hypothesis and data by choosing multiple endpoints, different cell lines and pectin solutions.

Based on the WHO 2020, the citation provided by the authors, the leading cause of cancer death worldwide is lung cancer and colorectal cancer follows that as the second cause of cancer death. Therefore, recommend correction of this statement on the Simple Summary, Abstract and Introduction sections.

Line 61. Recommend softening the language on the statement that concomitant combination of surgery and chemotherapy constitutes the standard treatment option for CRC. This is true that in curative management of locally advanced colon or rectal tumors, oxaliplatin based chemotherapy is used in combination with surgery. However, in metastatic CRC where irinotecan is used, the role of surgery is less clear and a recent randomized trial from Japan (JCOG1007 by Kanemitsu et al. JCO 2021) revealed no survival benefit from primary tumor resection.

Figure 1C and 1D: any explanation as to why SN38 did not impact the cell viability in the no pectin culture media? Is it in keeping with prior pre-clinical studies of SN38?

Although the authors have pointed out their statistical comparisons in the footnotes on the Figures, nevertheless it would be reasonable to highlight some of them in the text, particularly when based on those assessments an important conclusion is made.  For instance, in Line 277 it is pointed out that PC exhibited greater cytotoxic activity as compared to SolC, but it is unclear whether this reached statistical significance. The same for instance on Line 332, where a statement is made that in the case of PC, but not SolC, the number of cells undergoing apoptosis was much bigger. Another example is in Line 400, where the effect of PC in reducing the inflammatory markers is deemed to be higher than SolC, Was a statistical comparison made between the two SolC and PC treated LPS cells on either IL6 or COX2 concentrations?

A few minor grammar and spelling errors

Line 333 SN-38

Figure 3 footnote “Scheme 38”  

Line 396 use of either/or rather than both/and for COX2 and IL6.

Line 414 use of either/or rather than both/and for LF82 and K-12c600.

Line 449 “greater” cytotoxicity

 Line 500 “to” before “for” seems to be extra

Line 369, cytostatic or cytotoxic, since subG1 cellular fraction represents dead cells?

Line 395, recommend to make clarifications that pectins did not impact COX2 or IL6 levels in no-LPS treated cells.

Author Response

attached as a separate file

Reviewer 2 Report

The article by Anna Palko-Łabuz et. al. titled "Newly-obtained apple pectin as an adjunct to irinotecan therapy, presumptively reducing its side effects via influence on 3 colonic E. coli β-glucuronidase activity" describes apple pectin as a promising adjunct to Irinotecan therapy using colon cancer cell based assays.

Following is the major critique:

The title of the article immediately directs our attention towards the role of apple pectin on β-glucuronidase activity and how this helps in reducing toxicity due to Irinotecan. However, no direct data is shown by the authors that demonstrates apple pectin can reduce enzymatic transformation of SN-38G back to highly cytotoxic SN-38 in colon cancer cells.

Minor critiques:

1) While SN-38 alone does not reduce viability in the MTT and SRB assays as shown in Fig 1, it induces apoptosis in HCT116 cells as shown in Fig 2. It is not clear why that is. Also, authors have not discussed why HCT116 were more sensitive than CaCo2 cells.

2) The authors conclude that there is a synergistic interaction between SN-38 and apple pectin. However, it is difficult to conclude synergy or antagonism based on testing a single combination dose. Multiple doses and combinations are usually needed before arriving at a firm conclusion.

3) Although typically apoptosis can occur at any stage of the cell cycle, epithelial cells usually undergo apoptosis at the G1/S or G2/M phase of the cell cycle while Fig S2 shows that treatment with PC increased sub-GI cells whereas SN-38 increased G0/G1 cells. Moreover, it looks like PC induces a different mode of cell death compared to SN-38, and their combination behaves more like PC treatment i.e. increased sub-G1 cells. Does their combination potentially alter/interfere with SN-38's original mechanism?

Author Response

attached as a separate file

Round 2

Reviewer 2 Report

The authors have addressed the concerns which has significantly improved the manuscript.  Just one additional comment - Instead of 'Newly-obtained apple pectin', I might suggest 'Freshly extracted apple pectin', for consideration by the authors.